# Association between 24-Hour Movement Behaviors and Smartphone Addiction among Adolescents in Foshan City, Southern China: Compositional Data Analysis

**DOI:** 10.3390/ijerph19169942

**Published:** 2022-08-12

**Authors:** Zhiqiang Ren, Jianyi Tan, Baoying Huang, Jinqun Cheng, Yanhong Huang, Peng Xu, Xuanbi Fang, Hongjuan Li, Dongmei Zhang, Yanhui Gao

**Affiliations:** 1Department of Epidemiology and Biostatistics, School of Public Health, Guangdong Pharmaceutical University, Guangzhou 510315, China; 2Department of Medical Statistics, School of Basic Medicine and Public Health, Jinan University, Guangzhou 510632, China; 3School of Sport Science, Beijing Sport University, Beijing 100084, China; 4Department of Public Health Management and Social Medicine, School of Public Health, Guangdong Pharmaceutical University, No. 283, Jianghai Road, Haizhu District, Guangzhou 510315, China; 5Department of Epidemiology and Health Statistics, School of Basic Medicine and Public Health, Jinan University, No. 601, Huangpu Avenue West, Tianhe District, Guangzhou 510632, China

**Keywords:** 24-Hour movement behaviors, smartphone addiction, COVID-19, compositional analysis

## Abstract

Smartphone addiction has become a public health issue. To help reduce smartphone addiction, we assessed the combined effect of 24-Hour Movement Behaviors on smartphone addiction during Corona Virus Disease 2019 (COVID-19) home confinement in Foshan, China. Data were collected in a sample of 1323 senior middle school students ((mean age ± standard deviation): 16.4 ± 0.9 years; 43.46% males) during the COVID-19 lockdown. Their 24-Hour movement behaviors were assessed by a self-reported questionnaire, The Smartphone Addiction Scale-Short Version (SAS-SV). The compositional multiple linear regression model and compositional isotemporal substitution model were used to examine the association between the time budget composition of the day and smartphone addiction. Smartphone addiction occurred in 671 (50.72%) of the 1323 students. Compared with smartphone-addicted adolescents, non-smartphone-addicted adolescents had more moderate-to-vigorous physical activity (MVPA) and sleep duration (SLP), and less sedentary behavior (SB). The distribution of time spent in 24-Hour movement behaviors was significantly associated with smartphone addiction. The negative effect was found for the proportion of time spent in MVPA or SLP (*ilr*_1-MVPA_ = −0.453, *p* < 0.001. *ilr*_1-SLP_ = −3.641, *p* < 0.001, respectively) relative to the other three behaviors. Conversely, SB was positively associated with the score of smartphone addiction (*ilr*_1-SB_ = 2.641, *p* < 0.001). Reallocating one behavior to remaining behaviors was associated with smartphone addiction. Noticeably, the effects of one behavior replacing another behavior and of one behavior being displaced by another behavior were asymmetric. The 24-Hour movement behaviors of adolescents are closely related to smartphone addiction, and future intervention studies should focus on the compositional attribute of 24-Hour movement behaviors.

## 1. Introduction

It was well-established that a poor lifestyle pattern during adolescence is more likely to form in adulthood, which predisposes these adolescents to long-term adverse health outcomes. A recent systematic review showed that the proportion of Chinese students meeting the World Health Organization (WHO) recommendation that children/youth should participate in sufficient physical activity (PA) by engaging in moderate-to-vigorous physical activity (MVPA) for at least 60 min daily per week between 2005 and 2018 was about 31.1% [1]. In addition to PA, the problems of sedentary behavior (SB) and sleep duration (SLP) in elementary and middle school students are also in a serious situation [2,3,4]. A systematic review of 218 studies across 20 countries indicated that SLP has consistently declined over the past century [5]. A study showed that children and adolescents sit for 8.6 h on average, accounting for 62% of their waking time [6]. More importantly, a consistent body of literature has shown that the majority of smartphone use is for leisure and learning, and 90.9% of adolescents reported typically sitting as they used the smartphone [7].

According to a recent statistical report on Internet development in China, as of December 2020, the number of Internet users reached 989 million, and the proportion of Internet users using smartphones in China has reached 99.7%, of which 16.6% were primary and secondary school students under 19 years old [8]. While smartphones bring convenience to daily life, they also cause great harm on health among Chinese adolescents [9,10]. The use of a smartphone for a long time even results in a new kind of mental illness—smartphone addiction, also known as problematic smartphone use (PSU), which mainly refers to the physical, psychological or behavioral discomfort caused by the long-term use of the smartphone [11,12,13]. The latest research has found that smartphone addiction is related to a variety of health hazards, including psychosocial disorders such as anxiety and depression [14], traffic accidents and fatal injuries [15,16], as well as family and peer problems [17,18].

During the pandemic caused by Coronavirus Disease 2019 (COVID-19) and the lockdown established, people have reduced considerably their mobility and motor activity, which has led to an increase in unhealthy lifestyle habits, raising the risk of suffering from diseases [19]. People can only acquire the epidemic situation outside through TV, smartphones and other media. The survey indicated significant decreased time spent on physical activity, longer screen time and abnormal sleeping duration among Chinese school students compared with 3 months before the outbreak of COVID-19 [20]. Therefore, it is of great public health significance to comprehensively and systematically study the relationship between daily behaviors and smartphone addiction during the epidemic.

With regard to smartphone addiction, most research currently focuses on college students, but senior middle school students cannot be ignored because adolescence represents a critical period of development during which individual lifestyle choices and behavior patterns is cultivated. Additionally, research on smartphone addiction has mainly focused on psychological factors such as self-control, and anxiety or depression symptoms [14,21,22]. Recently, studies have begun to explore the relationship between PA and smartphone addiction, such as the relationship between the average daily steps, calorie consumption and smartphone addiction [23,24]. However, most studies regarded PA, SB and sleep as isolated factors to explore the relationship with health outcomes. Few researchers have studied these factors as a whole entity at the same time. In recent years, with the continuous development of the connotation of physical activity research, some scholars have put forward the concept of “24 h movement continuum”, which refers to the continuum of daily movement from no intensity to high intensity, including MVPA and light-intensity physical activity (LPA), SB and SLP [25,26]. Novel analytic approaches for physical activity surveillance such as Compositional Data Analysis (CoDA) make it possible to account for the constrained nature of daily movement behaviors during a 24 h day (24-Hour movement behaviors) [27,28]. It is more valuable to explore how to scientifically and reasonably allocate the time of each part of 24-Hour movement behaviors. The CoDA can quantify the health benefits caused by the substitution of 24-Hour movement behaviors from a group perspective, clarify the time allocation mode of each behavior that is most closely related to adverse health outcomes, accurately locate the best time balance point to maximize health benefits, and provide a valuable quantitative basis for public health policy-making. In order to comprehensively consider the relationship between PA, SB, and SLP and smartphone addiction, this study focuses on the following three questions based on CoDA: (1) Are there differences in 24-Hour movement behavior patterns of smartphone-addicted and non-smartphone-addicted students? (2) Is there a correlation between the 24-Hour movement behaviors and smartphone addiction? (3) How will the “dose-effect” relationship of the score of smartphone addiction change if one behavior is temporarily replacing another behavior for a fixed duration?

## 2. Materials and Methods

### 2.1. Participants

From 19 April to 25 April 2020, a survey was conducted among senior middle school students from Foshan city located in southern China. Their participation in our survey was voluntary, and they were free to withdraw at any time without being forced to complete the tasks. Electronic informed consent was obtained from each participant before the investigation, and then all participants were asked to answer self-rating questionnaires using the Wenjuanxing Online Survey System (https://www.wjx.cn/, accessed on 26 April 2020). Finally, a total of 1415 students were then recruited. Of them, 92 were excluded due to incomplete responses. Thus, only 1323 subjects were eligible for the final analysis, resulting in an effective response rate of 93.50%.

### 2.2. 24-Hour Movement Behaviors Assessment

The types of physical activity and metabolic equivalent (MET) intensities were assessed with a reference [29]. The questionnaire was adapted according to the physical activity characteristics of adolescents in China, and finally the questionnaire on physical activity behavior of adolescents was formed. Participants were asked to report the average time everyday spent the past week in moderate-intensity physical activities (dancing, tai chi, yoga, brisk walking/jogging, stairs, heavy housework, etc.) and vigorous-intensity physical activities (running, mountain climbing, swimming, big ball games, carrying heavy objects, aerobics, etc.). Since MVPA has a significant effect on promoting health, both moderate-intensity physical activities and vigorous-intensity physical activities combined into MVPA. SLP was considered the sum of the duration reported for their daily night sleep time and nap time. Average sedentary time was assessed with duration and types of activities in sitting, reclining or lying positions (using smartphones, tablets and other electronic devices, watching TV, playing cards, playing mahjong and other leisure entertainment, chatting with family and friends, reading and studying, etc.). LPA was determined by subtracting SLP, MVPA, SB from 24 h. Time spent in the four movement behaviors were summed and normalized to the proportion of the total time, which summed to 1.

### 2.3. Smartphone Addiction Assessment

The Smartphone Addiction Scale-Short Version (SAS-SV) was employed to measure the severity of smartphone addiction [30]. This self-rating scale contains 10 items. Each item is rated on a six-point Likert-type scale ranging from “1 = Strongly disagree” to “6 = Strongly agree” to reflect smartphone usage during the past month. Higher scores represent a higher risk of smartphone addiction. The SAS-SV score can be divided into “smartphone addiction” and “non-smartphone addiction” based on the recommended cut-off value (male 31; female 33). The Chinese version of the SAS-SV has been confirmed to have good reliability and validity [31]. The Cronbach’s alpha in our study was 0.91.

### 2.4. Other Variables Assessment

Respondents reported their anxious status (categorized into “no anxiety”, “mild anxiety”, “moderate anxiety”, and “severe anxiety”) [32] and regular exercise habit before the epidemic (“Yes,” and “No”) [33]. We also collected data on sex, age, height, weight and grade (“10th,” “11th” and “12th”). Information on indicators of parents, including education level (“Junior middle school and below”, “Senior middle school”, and “College or higher”), “Are parents working on the front line of the epidemic?” (“Yes”, and “No”) were also collected.

### 2.5. Statistical Analysis

Statistical analyses were performed using the SAS version 9.4 and R statistical system version 3.6.1. Descriptive statistics were calculated using means ± SD for continuous variables, frequency (percentage) for categorical variables. The outcome variable was tested to be approximately in line with the normal distribution. The t-test was used to compare the differences of the score of smartphone addiction on different demographic variables. Statistical significance was set a priori at *p* < 0.05.

Our analyses followed the guide to CoDA for physical activity, sedentary behavior and sleep research published by Chastin and colleagues [34]. Compositional descriptive statistics were calculated including compositional geometric means and variation matrix for 24-Hour movement behaviors. The compositional geometric mean is a measure of central tendency and was derived by calculating the geometric mean of the time spent in each movement behavior after the behaviors had been normalized to the proportion of the total time. The variation matrix is a measure of dispersion and was derived by calculating the variances of the logarithms of all possible pair-wise ratios (e.g., variance of ln (MVPA/SB)). Values closer to 1 indicated lower codependence and values closer to zero indicated higher codependence.

In order to appreciate the co-dependence between behaviors and emphasize the relative differences between subgroups of interest, compositional geometric mean bar plots were also calculated to display the relative movement behavior profiles for smartphone-addicted and non-smartphone-addicted adolescents. For each movement behavior, a log contrast was calculated between the compositional geometric mean of the entire sample and the compositional geometric mean of the health indicator subgroup (e.g., ln (SB subgroup/SB total sample)).

The constant sum constraint makes the proportions of time spent in SLP, SB, LPA and MVPA perfectly collinear. This can be overcome by using isometric log-ratio (ilr) data transformation [35]. Briefly, compositional data analysis involves expressing the daily movement behaviors in relative terms, as a set of isometric log-ratio coordinates. In this multiple linear regression model with the score of smartphone addiction as outcome, a specific isometric log-ratio transformation was used so that the first coordinate had MVPA as its numerator and the geometric mean of the remaining behaviors (SLP, SB and LPA) as its denominator. The first coordinate therefore contained all information regarding MVPA relative to the remaining behaviors. This enabled interpretation of the first regression coefficient from the model as the smartphone addiction association of MVPA relative to the remaining behaviors. The analysis was repeated for SLP, SB, LPA and MVPA as the first coordinate. The models were checked for linearity, normality, homoscedasticity and outliers to ensure assumptions were not violated.

Compositional isotemporal substitution used multiple linear regression models with the 24 h movement behaviors (expressed as ilr coordinates) and sociodemographic covariates as explanatory variables or predictors [36]. Daily movement behaviors were iteratively changed from the baseline composition (the compositional mean) to represent incremental 1-min increase/decrease in one behavior (e.g., MVPA) while the other behaviors (e.g., SB) were relatively decreased/increased to maintain a total daily maximum of 24 h. This process was repeated for each behavior. Subsequently, the differences in the estimated scores of smartphone addiction between the baseline (predicted mean) composition and the new compositions were calculated and plotted to aid interpretation.

## 3. Results

Of the 1323 eligible students for this study, 575 (43.46%) were boys, as shown in Table 1. The score of smartphone addiction significantly differed by age (*F* = 6.67, *p* = <0.001). Before the epidemic, there were 739 (55.86%) students with a regular exercise habit. Students who had regular exercise habits before the epidemic showed lower smartphone addiction scores than those who had not regular exercise habits (*F =* 8.68, *p =* 0.003). In addition, as anxiety degrees increased, students were more likely to have higher scores of smartphone addiction (*F* = 61.83, *p* = <0.001). During the epidemic, a total of 671 (50.72%) of the 1323 students were smartphone-addicted and the average score for smartphone addiction was 39.14 ± 6.68.

The geometric/arithmetic means for the percentage of time spent in each movement behavior in the 24 h period, and the minutes/day for each movement behavior after normalizing them to 1440 min/day, are presented in Table 2. In the total sample, the geometric means equate to participants spending 36.44% of the 24 h period in sleep, 37.21% in sedentary time, 25.73% in LPA, and 0.62% in MVPA. The variability of the data is summarized in the compositional variation matrix (Table 3). The smallest variances were observed for SLP and SB, for SLP and LPA, and for SB and LPA. The highest variances were observed for MVPA, which demonstrated that time spent in MVPA was the least co-dependent on the other behaviors. The compositional variation matrix of smartphone addicted and non-smartphone-addicted students is similar to that of the total sample.

Compositional geometric mean bar plot comparing the compositional mean of the entire sample with the compositional mean of group for all movement behaviors is shown in Figure 1. The proportion of time spent MVPA was dramatically higher and the proportion of time spent in SB was lower in the non-smartphone-addicted group relative to the smartphone-addicted group.

The association between movement behaviors and smartphone addiction relative to the other movement behaviors is displayed in Table 4. The composition of 24-Hour movement behaviors as a whole was significantly associated with smartphone addiction. After adjustments for covariates, time spent in SLP relative to other movement behaviors was negatively associated with smartphone addiction (*ilr_1-_*_SLP_ = −3.641; *p* = <0.001). Time spent in SB relative to other movement behaviors was positively associated with smartphone addiction (*ilr_1-_*_SB_ = 2.641; *p* = <0.001). Time spent in LPA relative to other movement behaviors was positively associated with smartphone addiction (*ilr_1-_*_LPA_ = 1.454; *p* = <0.001). Time spent in MVPA relative to other movement behaviors was negatively associated with smartphone addiction (*ilr_1-_*_MVPA_ = −0.453; *p* = <0.001).

Figure 2 shows the results of the compositional isotemporal reallocations based on adjusted models estimates. A range of time durations (from 1 to 8 min in 1-min increments) was reallocated to create an isotemporal reallocation plot for all possible pair-wise movement behaviors. Since the time of compositional isotemporal substitution cannot exceed the geometric mean, only maximal 8 min (G = 8.95) of isotemporal substitution effect results were given. Favorable changes in outcomes were observed when reallocating time to MVPA or SLP (i.e., showing a reduction). Detrimental associations were observed when time was reallocated to SB or LPA. For example, reallocating 5 min from SB to MVPA was associated with a lower score of −0.27 (95% CI = −0.41: −0.12). However, when SB or LPA replaced MVPA, the score of smartphone addiction increased.

## 4. Discussion

This study mainly explored the correlation between 24-Hour movement behaviors and smartphone addiction as well as the isotemporal substitution effect base on CoDA. To our knowledge, this is the first study to use compositional analyses to examine this relationship in adolescents. In terms of statistical methods, 24 h movement behaviors are treated as compositional data for analysis, which overcomes the difficulties of pseudo-correlation and collinearity so as to provide statistically more reasonable and comprehensive results.

Senior middle school students are in a critical and active period of physical and mental development, with strong curiosity but relatively weak self-control, and they are prone to smartphone addiction [37]. The degree of smartphone addiction showed a decreasing trend with age or grade, which is consistent with the result of a previous survey [38]. Younger students may be vulnerable to smartphone overuse or addiction. This may be due to the poor self-control caused by psychological maturity of adolescents, and it is also the result of the increased academic pressure of senior middle school students, which is to say, they are about to face the college entrance examination. At the same time, adolescents with regular exercise habits before the epidemic showed a lower tendency to smartphone addiction. On the one hand, physical fitness and tolerance will be improved in the process of regular exercise, and it can effectively alleviate the negative emotions caused by smartphone addiction, such as craving and withdrawal symptoms [39]. On the other hand, it can bring good psychological benefits to individuals, promote the development of mental health, and relieve negative emotions such as loneliness, depression and anxiety [40]. The result also showed that anxiety has a correlation with the score of smartphone addiction, which is consistent with the results of Kadir and Elhai [41,42]. These psychological factors even may act as mediators or modifiers to indirectly affect smartphone addiction [43,44].

The vast majority of people worldwide have been impacted by COVID-19. A wide variety of behaviors, including increased smartphone usage, decreased physical activity, more sedentary behavior, anxiety, and depression, were changed compared with prior academic years [45]. In our study, the detection rate of smartphone addiction was 50.72%, which is higher than that before the epidemic [7,46]. Compared with non-smartphone-addicted adolescents, smartphone-addicted adolescents had more SB, and less MVPA and SLP. During the quarantine, adolescents spent more time on SB, and this was likely to increase the use of digital devices. The increase in digital devices’ use before going to bed affected sleep latency and wake time. A previous survey showed that there is an association between higher levels of physical activity and improved sleep quality [47]. The non-smartphone-addicted adolescents’ behavior pattern is exactly in accord with the most ideal combinations of movement behaviors (e.g., high sleep, low sedentary behavior, high physical activity) that may be important for optimal health in the early years [48].

Movement behaviors as a structural entity were closely related to smartphone addiction. It can be seen that there was a significant negative correlation between MVPA and smartphone addiction. This result was consistent with the research of Yang Guan [24]. Similarly, physically inactive adolescent students were more likely to present with problematic smartphone use than those who were physically active [49]. At the same time, there was a significant negative correlation between sleep and smartphone addiction. Many studies showed that there may be a bidirectional causal relationship between sleep and smartphone addiction. Using smartphones for a long time before going to bed weakens the motivation to fall asleep, and provides an important opportunity and psychological basis for revenge staying up late, leading to sleep difficulties and less sleep duration [50]. On the contrary, sleep was significantly associated with the degree of smartphone dependence [51,52], and it can also indirectly affect smartphone addiction through factors such as depression [53]. Our finding showed that SB is positively correlated with smartphone addiction, suggesting that individuals who allocated more time for daily sitting used smartphones for greater periods. Some surveys found that smartphone use was positively associated with SB during the weekdays and weekends [7,54]. They indicated that smartphones, despite their portability and mobility, are primarily sedentary devices for all individuals. Sedentary behavior is closely related to traditional forms of screen-based activities (e.g., watching television, playing video games and surfing the Internet), which inevitably leads to excessive sedentary behavior for adolescents who get addicted to smartphones or computers. There was very little literature on the association between LPA and smartphone addiction.

We found significant associations between smartphone addiction and movement behaviors with reallocation of different times. Virtually, this result was in accordance with the compositional multiple linear models. Nationally, between 2016 and 2017, 35–37% of children and adolescents reported spending more than 2 h per day on electronic screens [55]. The 2018 Physical Activity Guidelines for Children and Adolescents in China put forward the recommended amount of daily physical activity and sedentary activity for children and adolescents aged 6–17 for the first time. However, statements such as “get at least 30 min of MVPA five times per week” or “reduce time spent in SB” are in this sense incomplete, since there are no recommendations about how the rest of the day should be accommodated. As research continues, new paradigms related to daily activity and health are emerging. For example, the Canadian 24-Hour Movement Guidelines have been proposed that integrate the balance of time spent per day in light to vigorous-intensity physical activity, SB and sleep to enhance health outcomes in those aged 5–17 years [56]. Lately, 24 h activity guidelines have been released in Australia [57], New Zealand [58], South Africa [59], Finland [60], Croatia [61], and the WHO [62]. In contrast, messages such as “sit less, move more” are consistent with the time-use epidemiology paradigm, as they are explicit appeals to swap one behavior for another. Thus, it is critically important for public health and physical education agencies developing activity and health guidelines for China to fully recognize that compositional attribute of daily movement behaviors is fundamental for any implementation to be effective.

The estimated change in the scores of smartphone addiction cannot be directly explained as the effect of time redistribution from one movement behavior to another. The calculations for Figure 2 used the mean movement behavior composition as the baseline. However, the estimated change in the score of smartphone addiction will differ for reallocation at another starting composition. Due to the compositional nature of 24 h time-use data, the predictions for changes in the score of smartphone addiction related to reallocations between movement behaviors must be made with reference to a baseline or starting composition. The reallocation results should be interpreted as highlighting differences between groups with different activity patterns, rather than as indicative of what would necessarily happen if people changed their activity mix.

Consistent with other studies using compositional data analysis, we found that the effects of isotemporal substitution in Figure 2 were asymmetry [34,63]. For example, the estimated detrimental results of a reduction in MVPA were greater than the estimated favorable results for the inverse reallocation, that being an increase in MVPA time. Chastin et al. hypothesized this may be due to two isolated factors: first, that time spent in MVPA is much shorter than time spent SB, and therefore any reallocation of these behaviors constitutes substantially difference percentages of time for each behavior respectively; second, it is highlighted that deconditioning or weight gain requires lower stimuli (i.e., occurs more easily) than the equivalent conditioning or weight loss [34]. However, it is also possible that this asymmetric relationship is an artifact of the analysis, as 10 min of SB will affect the compositions to a lesser extent than 10 min of MVPA, which exactly reflects that the compositional data focuses on relative information rather than absolute information. Nevertheless, when energy expenditure is considered, theoretically, a 10 min reallocation to or from SB time will have the same size of effect on energy expenditure. Therefore, this asymmetric association needs further investigation.

The strength of the present study is the first to the authors’ knowledge to utilize compositional data analysis to model the association between movement behaviors and the score of smartphone addiction in adolescents, as well as to examine associations of reallocating time in different movement behaviors. In addition, there was high compliance with the study protocol, and the outcome was assessed with a commonly used and well-validated questionnaire. Despite these strengths, this study is not without limitations. This study uses online questionnaire surveys to record 24 h movement behaviors, which lacks objectivity compared to measurement methods such as accelerometers. The sample size was modest, and the characteristics of the sample were unrepresentative of the Chinese adolescent population. Data were collected from geographically limited areas, so the conclusions need further research. Last but not least, this study is a cross-sectional design that does not allow us to make inferences about causality between 24 h movement behaviors and smartphone addiction. The research may also have some residual confounding factors, such as loneliness, self-control and personality.

## 5. Conclusions

In conclusion, 24-h movement behaviors of adolescents are closely related to smartphone addiction during COVID-19. The behavior pattern with more MVPA and less SB has a positive effect on optimal health, and increasing MVPA and reducing SB is one of the effective methods to avoid smartphone addiction of adolescents. The CoDA provides a powerful means to understand the relationship between 24 h movement behaviors and outcomes. This study provides scientific and comprehensive theoretical guidance for the intervention research of smartphone addiction in adolescents, and provides enlightening ideas for related intervention practices. The opening of an overall research paradigm is conducive to exploring the optimal distribution of time spent in different movement behaviors throughout the day, and contributes to further development of comprehensive guidelines for physical activity, sedentary behavior and sleep. In addition, future research about smartphone addiction should focus on the compositional attribute of 24 h movement behaviors and use larger and representative samples for more meaningful and longitudinal research.

## Figures and Tables

**Figure 1 ijerph-19-09942-f001:**
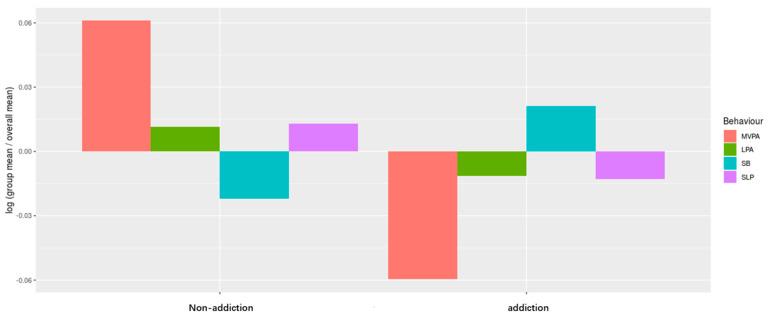
Comparison of 24-Hour movement behaviors of participants by group. SLP, sleep duration; SB, physical activity; LPA, light-intensity physical activity; MVPA, moderate-intensity physical activity.

**Figure 2 ijerph-19-09942-f002:**
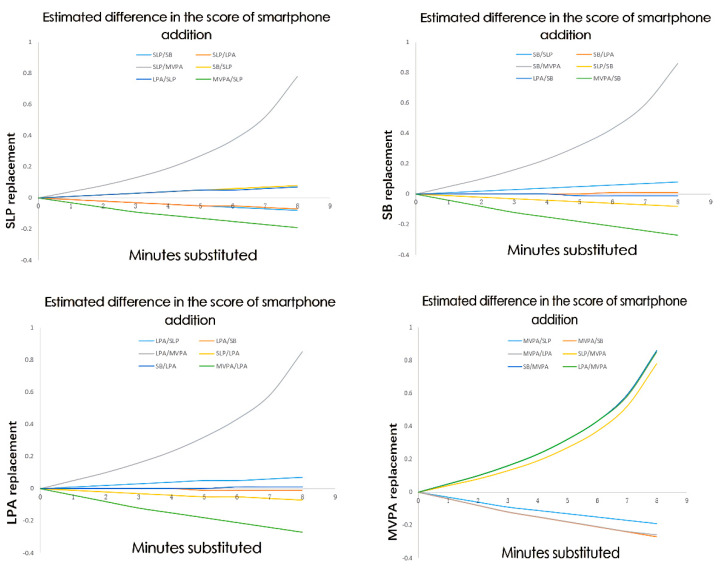
Changes in the scores of smartphone addiction after one activity replacing other behaviors at the different fixed durations of time. LPA, light-intensity physical activity; MVPA, moderate-to-vigorous-intensity physical activity; SB, sedentary time; SLP, sleep duration. X-axis represents the number of minutes substituted. Y-axis represents the estimated difference in the score of smartphone addiction. All models are adjusted for age, sex, grade, only child or not, BMI, regular exercise, anxiety, parents’ education level, and whether work on the front line of the epidemic or not.

**Table 1 ijerph-19-09942-t001:** Socio-demographic characteristics of participants during the epidemic.

Variables	N (%)	SAS-SV Scores (Mean ± SD)	*F*	*p*
Sex			1.08	0.299
Boy	575 (43.46)	31.49 ± 10.81		
Girl	748 (56.54)	32.06 ± 8.87		
Age			6.76	<0.001
15 years	250 (18.90)	33.10 ± 9.18		
16 years	494 (37.34)	32.35 ± 10.05		
17 years	399 (30.16)	31.56 ± 9.34		
>17 years	180 (13.61)	29.10 ± 10.20		
Grade			15.92	<0.001
10th	553 (41.80)	33.09 ± 9.67		
11th	500 (37.79)	31.89 ± 9.73		
12th	270 (20.41)	29.05 ± 9.47		
Only child or not			0.58	0.447
Yes	495 (37.41)	31.55 ± 9.50		
No	828 (62.59)	31.97 ± 9.92		
BMI			1.11	0.345
Underweight	362 (27.36)	32.37 ± 9.62		
Normal weight	745 (56.31)	31.72 ± 9.87		
Overweight	109 (8.24)	31.94 ± 9.05		
Obese	107 (8.09)	30.46 ± 10.15		
Had regular exercise before the epidemic			8.68	0.003
Yes	739 (55.86)	31.11 ± 9.97		
No	584 (44.14)	32.70 ± 9.42		
Father’s education level			1.66	0.175
Junior high school and below	455 (34.39)	32.35 ± 9.71		
High school	491 (37.11)	31.83 ± 9.78		
College or higher	377 (28.50)	31.14 ± 9.73		
Mother’s education level			1.17	0.319
Junior high school and below	556 (41.03)	32.17 ± 9.78		
High school	465 (35.15)	31.55 ± 9.79		
College	302 (22.82)	31.55 ± 9.53		
Parents working on the front line of the epidemic?			3.07	0.080
Yes	68 (5.14)	29.79 ± 9.32		
No	1255 (94.86)	31.92 ± 9.78		
Anxiety			61.83	<0.001
No anxiety	1015 (76.72)	30.13 ± 9.03		
Mild anxiety	222 (16.78)	35.37 ± 8.52		
Moderate anxiety	56 (4.23)	41.70 ± 10.66		
Severe anxiety	30 (2.27)	43.90 ± 13.70		
Smartphone addiction			10.65	<0.001
Addiction	671 (50.72)	39.14 ± 6.68		
Non-addiction	652 (49.28)	24.27 ± 5.94		

**Table 2 ijerph-19-09942-t002:** The 24-Hour movement behavior’s central tendency of the participants in minute/day (% of 24 h).

	Geometric Mean			Arithmetic Mean (min)		
	Addiction	Non-Addiction	Total	Addiction	Non-Addiction	Total
SLP	517.82 (33.36)	531.65 (36.44)	524.74 (36.44)	479.14 (33.27)	481.61 (33.45)	480.36 (33.36)
SB	547.34 (37.04)	524.16 (37.21)	535.82 (37.21)	546.52 (37.95)	519.77 (36.10)	533.34 (37.04)
LPA	366.34 (28.08)	374.69 (25.73)	370.51 (25.73)	394.32 (27.38)	414.60 (28.79)	404.32 (28.08)
MVPA	8.50 (1.53)	9.50 (0.62)	8.93 (0.62)	20.02 (1.39)	24.02 (1.67)	21.98 (1.53)

SLP, sleep duration; SB, physical activity; LPA, light-intensity physical activity; MVPA, moderate-intensity physical activity. Movement behaviors have been normalized to 1440 min.

**Table 3 ijerph-19-09942-t003:** Compositional variation matrix of time spent in SB, SLP, LPA and MVPA in minutes/day.

	SLP	SB	LPA	MVPA
Total				
SLP	0.00	0.14	0.21	1.24
SB	0.14	0.00	0.56	1.70
LPA	0.21	0.56	0.00	1.74
MVPA	1.24	1.70	1.74	0.00
Addiction				
SLP	0.00	0.13	0.22	1.31
SB	0.13	0.00	0.58	1.67
LPA	0.22	0.58	0.00	1.74
MVPA	1.31	1.67	1.74	0.00
Non-addiction				
SLP	0.00	0.16	0.21	1.15
SB	0.16	0.00	0.58	1.57
LPA	0.21	0.58	0.00	1.84
MVPA	1.15	1.57	1.84	0.00

SLP, sleep duration; SB, physical activity; LPA, light-intensity physical activity; MVPA, moderate-intensity physical activity.

**Table 4 ijerph-19-09942-t004:** Correlation between 24-Hour movement behavior and smartphone addiction score during the epidemic.

*ilr* Coordinates	$\hat{\beta }$	*SE*	*t*	*p*	*Standardized* $\hat{\beta }$
*ilr_1-_* _SLP_	−3.641	0.755	−4.821	<0.001	−0.416
*ilr_1-_* _SB_	2.641	0.601	4.395	<0.001	0.351
*ilr_1-_* _LPA_	1.454	0.350	4.154	<0.001	0.125
*ilr* * _1-_ * _MVPA_	−0.453	0.341	−3.460	<0.001	−0.105

*ilr_1-_*_SLP_, SLP as the first coordinate; *ilr_1-_*_SB_, SB as the first coordinate; *ilr_1-_*_LPA_, LPA as the first coordinate; *ilr1_-_*_MVPA_, MVPA as the first coordinate. Regression coefficients correspond to change in the log-ratio of the given behavior compared to the other behaviors. All models are adjusted for age, sex, grade, only child or not, BMI, regular exercise, anxiety, parents’ education level, and whether work on the front line of the epidemic or not.

## Data Availability

All participants provided written informed consent before participating in the study.

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
