# Peer review of "Association between 24-Hour Movement Behaviors and Smartphone Addiction among Adolescents in Foshan City, Southern China: Compositional Data Analysis"

_ijerph, 2022, doi:10.3390/ijerph19169942_

Round 1

Reviewer 1 Report

I have reviewed this manuscript, which presents a very precise question about the relations between the 24-hour Movement Behaviours and Smartphone Addiction in Adolescents 

Regarding the abstract, I would recommend 

-standardize the font size,

-using the same criteria by writing each acronym and what it stands for the first time they are mentioned in the article.

-Since the data are collected from a geographical limited area, this fact should be mentioned in the abstract and in the title. The authors came up with universal results, despite the reduced area of applicability of data. This should be explained not only in the abstract, but in the discussion and the conclusions. 

The approach used to describe the problem is very systematic and therefore adequate. Materials and Methods (paragraph again written with different font sizes) are very clear in respect to content, and thoroughly researched. They present valuable information about the matter, supported by the final bibliography. 

The expression “a survey was conducted among high school students in a middle school” sounds odd to me. Maybe the differences among educational systems make advisable to establish the grades and ages not only in the tables, but in the paragraph 2.1. “Participants”

Tables and figures define the research carried out in an adroit manner.

The bibliography covers all necessary references. 

There are some minor problems of language. 

The conclusions deserve more development: is the area of Foshan city determining the outcomes in some way? Its weather? Its prefecture or province laws?  Please, clarify this. 

I think the third line of Abbreviations contains a typo. Check it, please. 

In summary, as the size of the sample is modest, I think the title and the abstract should reflect to what extent the results are or aren’t universal. Please, review the differences of font size in the text. I expect my comments would help to strengthen this article and make it more readable.

Author Response

Response to Reviewer 1 Comments

Re: IJERPH-1830554 entitled ‘Association between 24-hour Movement Behaviors and Smartphone Addiction among Adolescents in Foshan city, Southern China: Compositional Data Analysis.’

Dear reviewer 1,

Thank you for the suggestions and constructive comments. We have revised the manuscript responding to all your suggestions accordingly. The comments you provided helped us to clarify and improve the manuscript. For each comment, please find our point-by-point responses below. As required, we provided one version of our resubmitted manuscript with changes highlighted in yellow. Thanks again.

Kind regards,

Yanhui Gao

Department of Public Health and Preventive Medicine,

School of Basic Medicine and Public Health, Jinan University,

No. 601, Huangpu Avenue West, Tianhe District, Guangzhou, 510632, China.

Point 1: Regarding the abstract, I would recommend standardize the font size.

Response 1: Thank you for your comments. We have standardized the font size of the abstract and full text, as well as materials and methods.

Point 2: Regarding the abstract, I would recommend using the same criteria by writing each acronym and what it stands for the first time they are mentioned in the article.

Response 2: Thank you for your comments. We used the same criteria by writing each acronym and what it stands for the first time they are mentioned in the article in the revised manuscript.

Point 3: Regarding the abstract, I would recommend since the data are collected from a geographical limited area, this fact should be mentioned in the abstract and in the title. The authors came up with universal results, despite the reduced area of applicability of data. This should be explained not only in the abstract, but in the discussion and the conclusions.

Response 3: Thank you for your comments. According to your suggestion, we have revised the abstract and title. At the same time, more explanations have been given in the discussion and conclusions.

Point 4: The expression “a survey was conducted among high school students in a middle school” sounds odd to me.

Response 4: Thank you for your comments. We have revised it to “a survey was conducted among senior middle school students from Foshan city located in southern China.”, and gave a unified description of the full text to avoid this ambiguity in the revised manuscript.

Point 5: The conclusions deserve more development: is the area of Foshan city determining the outcomes in some way? Its weather? Its prefecture or province laws?  Please, clarify this.

Response 5: Thank you for your comments. Foshan city is an ordinary city in southern China. There is currently no sufficient evidence to show that the relationship between adolescents' movement behaviors or daily activities and the smartphone addition might be influenced by the study area. In addition, the weather was in spring during the survey period, and outdoor sports or exercise activities were not restricted. On the contrary, during the pandemic caused by COVID-19 and study period, China implemented a unified epidemic prevention and control policy, and there was no difference between Foshan and other regions. people have to change their daily habits and reduced their mobility. This may lead to decreased physical activity and increased sedentary behavior of adolescents, which in turn can lead to increased levels of mobile phone addiction. However, this study is a cross-sectional study, we did not collect information on the movement behaviors and smartphone addiction of the participants before the epidemic, and it is impossible to compare before and after the epidemic. According to the existing literature, we can confirm that physical activity and sedentary behavior are related to mobile phone addiction. However, previous studies often ignored the attribute of component data. This study can provide a more comprehensive and systematic study of the relationship between daily behavior and smartphone addiction during the epidemic period. It can put forward specific life behavior suggestions for the long-term isolated population during the epidemic period, which has important public health significance. These are described in paragraphs 3 and 8 of the discussion.

Point 6: I think the third line of Abbreviations contains a typo.

Response 6: Thank you for your comments. We have corrected this typo.

ilr: sometric log-ratio → ilr: isometric log-ratio

Point 7: In summary, as the size of the sample is modest, I think the title and the abstract should reflect to what extent the results are or aren’t universal.

Response 7: Thank you for your comments. Based on your suggestion, the title and abstract have been revised in the resubmitted manuscript, and the extrapolation of the conclusions is discussed.

Reviewer 2 Report

Dear authors, congratulations on the study you have carried out. The introduction is brief but very correct and adequate to know the state of the subject of the study. 

I would recommend that you write the three research questions as what they are, questions and not phrases in the affirmative.

The method is well laid out and the instruments are correctly referenced. 

The results are presented with very clear and well-designed tables and graphs. This is appreciated.

The discussion is a very important aspect of this study and they have been able to elaborate it in a thorough way. I think that if the three research questions were correctly posed before the method, the discussion would clarify what is being answered with the comments and data provided.

A final section on the theoretical and practical implications of the conclusions drawn from the study is missing.

Thank you very much.

Best regards.

Author Response

Response to Reviewer 2 Comments

Re: IJERPH-1830554 entitled ‘Association between 24-hour Movement Behaviors and Smartphone Addiction among Adolescents in Foshan city, Southern China: Compositional Data Analysis.’

Dear reviewer 2,

Thank you for the suggestions and constructive comments. We have revised the manuscript responding to all your suggestions accordingly. The comments you provided helped us to clarify and improve the manuscript. For each comment, please find our point-by-point responses below. As required, we provided one version of our resubmitted manuscript with changes highlighted in yellow. Thanks again.

Kind regards,

Yanhui Gao

Department of Public Health and Preventive Medicine,

School of Basic Medicine and Public Health, Jinan University,

No. 601, Huangpu Avenue West, Tianhe District, Guangzhou, 510632, China.

Point 1: I would recommend that you write the three research questions as what they are questions and not phrases in the affirmative.

Response 1: Thank you for your comments. According to your suggestion, we have rewritten them into three research questions in the revised manuscript.

 1) Are there differences in 24-hour movement behavior patterns of smartphone addicted and non-smartphone addicted students?

2) Whether there is a correlation between the 24-hour movement behaviors and smartphone addiction;

3) How will the “dose-effect” relationship of the score of smartphone addiction change if one behavior isotemporaly replacing another behavior for a fixed duration.

Point 2: I think that if the three research questions were correctly posed before the method, the discussion would clarify what is being answered with the comments and data provided.

Response 2: Thank you for your comments. According to your suggestion, we have further clarified the three research questions in the discussion section. In response to the first question, we supplemented and pointed out the differences in 24-hour movement behaviors between smartphone addicted and non-smartphone addicted adolescents in paragraph 3 of the discussion part, and explained the possible reasons.

  Compared with non-smartphone addicted adolescents, smartphone addicted adolescents had more SB, and less MVPA, SLP. During the quarantine, adolescent spent more time on SB, and this was likely to increase the use of digital devices. The increase in digital devices use before going to bed affected sleep latency and wake time. Previous survey showed that there is an association between higher levels of physical activity and improved sleep quality.

In response to the second and third question, we clarified the three research questions paragraph 4, 5 and 6 of the discussion part.

Point 3: A final section on the theoretical and practical implications of the conclusions drawn from the study is missing.

Response 3: Thank you for your comments. According to your suggestion, we have supplemented the theoretical and practical implications of the conclusions drawn from the study in final section.

The behavior pattern with more MVPA and less SB has a positive effect on optimal health, and increasing MVPA and reducing SB is one of the effective methods to avoid smartphone addiction level of adolescents.

The opening of an overall research paradigm is conducive to exploring the optimal distribution of time spent in different movement behaviors throughout the day, and is contributive to further development of comprehensive guidelines for physical activity, sedentary behavior and sleep. In addition, future research about smartphone addiction should focus on the compositional attribute of 24-hour movement behaviors and use larger and representative samples for more meaningful and longitudinal research.
